# Eight Weeks of Exercising on Sand Has Positive Effects on Biomechanics of Walking and Muscle Activities in Individuals with Pronated Feet: A Randomized Double-Blinded Controlled Trial

**DOI:** 10.3390/sports10050070

**Published:** 2022-05-02

**Authors:** Amir Ali Jafarnezhadgero, Amir Fatollahi, Urs Granacher

**Affiliations:** 1Department of Sport Managements and Biomechanics, Faculty of Educational Science and Psychology, University of Mohaghegh Ardabili, Ardabil 56131-56491, Iran; amiralijafarnezhad@gmail.com (A.A.J.); amiraf14618@gmail.com (A.F.); 2Division of Training and Movement Sciences, Research Focus Cognition Sciences, University of Potsdam, 14469 Potsdam, Germany

**Keywords:** flat foot, *free moment*, gait, loading rate, training

## Abstract

This study aimed to investigate the effects of eight weeks of barefoot running exercise on sand versus control on measures of walking kinetics and muscle activities in individuals with diagnosed pronated feet. Sixty physically active male adults with pronated feet were randomly allocated into an intervention or a waiting control group. The intervention group conducted an 8-weeks progressive barefoot running exercise program on sand (e.g., short sprints) with three weekly sessions. Pre and post intervention, participants walked at a constant speed of 1.3 m/s ± 5% on a 18 m walkway with a force plate embedded in the middle of the walkway. Results showed significant group-by-time interactions for peak impact vertical and lateral ground reaction forces. Training but not control resulted in significantly lower peak impact vertical and lateral ground reaction forces. Significant group-by-time interactions were observed for vastus lateralis activity during the loading phase. Training-induced increases were found for the vastus lateralis in the intervention but not in the control group. This study revealed that the applied exercise program is a suitable means to absorb ground reaction forces (e.g., lower impact vertical and lateral peaks) and increase activities of selected lower limb muscles (e.g., vastus lateralis) when walking on stable ground.

## 1. Introduction

Foot pronation has previously been defined as a normal rolling movement localized at the subtalar joint of the foot [1]. More specifically, foot pronation primarily occurs during the first half of the walking stance phase [2] and is characterized by ankle eversion and dorsiflexion as well as abduction of the forefoot [2]. However, excessive foot pronation is related to lower limb malalignment and may cause musculoskeletal injuries [3]. In youth aged 2–16 years, prevalence rates for individuals with pronated feet (PF) range between 48% and 78% [4]. For adults aged 18–50 years, prevalence rates are in the range of 2–23% [5,6,7].

With regards to the biomechanics of walking, there is evidence that individuals with PF show a significantly lower second peak (propulsion phase) of vertical ground reaction force (GRF) [1], greater lateral-medial GRF [8], and greater [9] or similar [8] peak-*free moment* amplitudes. Of note, different researchers have observed that greater peak vertical GRF during heel contact and vertical loading rates are associated with walking and/or running-related injuries [8,10]. Pronation velocity has previously been used to assess running stability properties, and there seems to be an association between pronation velocity and running injuries such as patella femoral pain syndrome [11,12,13].

Free moment amplitudes have been introduced as a marker of lower limb torsional load [14,15], and greater *free moment* amplitudes may constitute a risk factor for tibial stress fractures in runners [8,14]. With regards to muscle activities, there is evidence that individuals with PF compared with age-matched healthy controls showed both, higher lower limb muscle activities (i.e., tibialis anterior/posterior, plantar flexors) as well as lower activities of evertor muscles (e.g., peroneus) [16]. Data from original research [17] indicated that participants with a history of lower limb overuse injuries such as stress fractures showed a greater subtalar pronation excursion. A recent systematic review and meta-analysis reported high effect sizes in the range of 0.78 to 1.52 for PF posture as a risk factor for developing the tibial stress fracture and low effect sizes (effect size = 0.28 to 0.33) for PF posture as a risk factor to suffer from patellofemoral pain [18]. Another systematic review [19] showed trivial to moderate effects (effect size = 0.10 to 0.61) for rearfoot kinematic variables as a risk factor for patellofemoral pain. A recent meta-analysis indicated that peak rearfoot eversion was the only significant risk factor associated with the development of lower limb tendinopathy [20]. Accordingly, Mousavi et al. [20] concluded that rearfoot kinematics and kinematic chain movements should be considered in the prevention and management of lower limb tendinopathy. Therefore, it is important to design and develop adequate training regimens for PF treatment.

Sand could be a promising means to be used in PF therapy because it is rather easy to access worldwide and cost-effective. Of note, sand is characterized by high shock absorption during the loading phase [21]. At the same time, the unstable surface poses great demands on the somatosensory system due to its unpredictable three-dimensional surface [21]. There is preliminary evidence that walking or running on sand has a significant impact on the kinetics and kinematics of locomotion [21,22]. In addition, a previous study has shown that exercise on sand compared with training on firm ground increases energy costs and reduces impact forces [23].

Of note, specific biomechanical and physiological characteristics are associated with exercising on sand [22,23]. For instance, walking on sand resulted in significant increases in hip flexion, knee flexion, and ankle dorsiflexion angles in multiple sclerosis patients during the swing phase of walking [21]. Another study demonstrated lower *free moment* amplitudes and vertical loading rates when walking on sand [22]. Accordingly, sand seems to absorb impact forces and may thus reduce muscle damage and soreness [24]. In a previous study, we evaluated the effects of long-term training on sand on lower limb muscle activities during running but not walking [25]. Of note, walking mechanics are different from running mechanics with regard to the magnitude of ground reaction forces and muscle activities [26,27,28]. Therefore, it is timely to examine the effects of long-term exercising on sand on walking kinetics and muscle activities since walking is a natural human activity. Therefore, this study aimed to investigate the effects of 8-weeks of barefoot running exercise on sand versus control on measures of walking kinetics and muscle activities in individuals with diagnosed pronated feet. We hypothesized that exercise on sand decreases GRF amplitudes, loading rates, and *free moment* amplitudes and increases muscle activities during walking [21,22].

## 2. Materials and Methods

This study was designed as a double-blinded randomized controlled trial (Figure 1). An envelope concealment method was used to allocate study participants. Participants and examiners were unaware of group allocation. In other words, participants and assessors were blinded.

The G*Power software was used to calculate an a priori power analysis with the F-test family using a related study as a reference that evaluated the kinetics of walking in male adults with PF [8]. An alpha level of 0.05, a type II error rate of 0.20, and an effect size of 0.80 for peak vertical GRF from the study of Jafarnezhadgero et al. [22] were used to compute the power analysis. Findings showed that at least 20 participants per group would be needed to receive a significant group-by-time interaction effect.

Sixty participants aged 18–26 years with diagnosed PF were recruited from a clinic in Ardabil City, Iran, and were randomly allocated into an intervention or a waiting control group. A kicking ball test indicated that all participants were right foot dominant [29]. Given that males and females differ with regard to their biomechanical characteristics during walking, we recruited males only [30]. To be included in this study, participants had to be male, show a navicular drop of >10 mm [22], rear foot eversion of >4° [31], and a Foot Posture Index of >10 [22]. An orthopedic specialist diagnosed the participants during static standing [32,33]. The static measurements of the calcaneal deviation and the medial arch angle were recorded while the participants stood barefoot and in an erect stance position on a level floor [31]. Calcaneal deviation was defined as the angle between the posterior midline of the calcaneus and perpendicular line to the level floor that was measured with a protractor. The Foot Posture Index includes six items in order to classify foot type [33,34]. A full definition of the Foot Posture Index has previously been reported in the literature [33,34]. As exclusion criteria, we defined: (i) history of trunk and/or lower limbs surgery, (ii) orthopedic conditions (except for PF), and (iii) greater than 5 mm limb length differences. Prior to the start of the study, the study procedures were described to all participants. Thereafter, written informed consent was obtained from the study participants. The study protocol was approved by the local ethics committee of the Ardabil Medical Sciences University (IR.ARUMS.REC.1398.484) and registered by the Iranian clinical trial organization (IRCT20191211045704N1). The study was conducted in agreement with the latest version of the Declaration of Helsinki. Appendix A (Table A1) illustrates the CONSORT 2010 checklist.

### 2.1. Experimental Set-Up and Data Processing

To collect GRF data during walking on stable ground, a force platform (Bertec Corporation, Columbus, OH, USA) was used and embedded in an 18-m walkway. During testing, subjects walked barefoot at a constant speed of ~1.3 m/s ± 5%. Speed of walking was monitored through two sets of infrared photocells. As previously described by Jafarnezhadgero et al. [35], kinetic data were sampled at 1000 Hz. GRFs were smoothed using a 20 Hz (4th order Butterworth filter, zero lag) low-pass filter. Heel contact and toe off were determined through the Bertec force plate. A 10 N threshold was used to determine the stance phase of the walking. The dependent variables extracted from GRF data include [35] first (FzHC) and second vertical peak force (FzPO), braking (FyHC) and propulsion forces (FyPO), and the lateral (FxHC) and medial GRF (FxPO) peaks. Peak GRF values were normalized using body weight (BW). The time period between the heel strike and the corresponding peak of FzHC was defined as TTP [8]. The slope between heel strike and FzHC on the vertical curve was defined as loading rate [8]. The *free moment* values were calculated as follows [8]:$$Free\;moment=M_{z}-\left(F_{y}\times CoP_{x}\right)+\left(F_{x}\times CoP_{y}\right)$$
where *M_z_* is defined as the moment around the vertical axis, *x* and *y* are medio-lateral and anterior-posterior components of the center of pressure (*CoP*), and *F_x_* and *F_y_* are the medio-lateral and anterior-posterior GRF components. Moreover, *free moment* values were normalized using BW × height. All dependent variables were averaged across five trials [35].

An electromyographic apparatus (Biometrics Ltd., Nine Mile Point Ind. Est, Newport, U.K.) with bipolar Ag/AgCl surface electrodes (25 mm center-to-center distance; input impedance of 100 MΩ; and common mode rejection ratio of >110 dB) was applied to record electromyographic data of the tibialis anterior, medial gastrocnemius, biceps femoris, semitendinosus, vastus lateralis, vastus medialis, rectus femoris, and gluteus medius muscles of the right limb [36]. Raw electromyographic data were recorded at a sampling rate of 1000 Hz. The skin surface was cleaned and shaved using alcohol to lower impedance in accordance with the European recommendations for surface electromyography (SENIAM). The stance phase was divided into the loading (0–20% of gait cycle), mid-stance (20–47% of gait cycle), and push-off (47–70% of gait cycle) sub-phases for electromyographic analyses [22]. Maximum voluntary isometric contraction (MVIC) was recorded using a handheld dynamometer along with an isometric belt (where the joint is locked) [37] for each muscle separately to normalize electromyographic amplitudes during walking to MVIC (Appendix B, Table A2). The applied normalization procedures were realized in accordance with recommendations from Besomi et al. [37]. For example, the participants were encouraged to perform the tests at maximal effort [37]. Three test trials were conducted with rest periods of 1–2 min in between [37]. This instrument is important to standardize the test during the performance of maximal contractions. The maximum value of the MVIC test was considered for normalization purposes [37].

### 2.2. Experimental Procedures

A 5 min warm-up protocol consisting of 2 min jogging followed by 3 min stretching was conducted with all participants prior to testing. Participants walked barefoot across the 18-m walkway at a constant speed of 1.3 m/s. Walking speed was monitored using electronic timing gates (Swift Performance Equipment, Wacol, New South Wales, Australia). Five successful test trials had to be performed. If the foot touched the middle of the force plate and if electromyography signals were artifact-free, a trial was considered successful. MVIC tests were performed after the walking trials for each muscle to normalize electromyography data (Appendix B, Table A2). Similar procedures were applied during pre and post testing.

The intervention group performed running exercises on sand over a period of 8 weeks with three sessions per week. The intervention program was progressively designed using walking, jogging, striding, bounding, and galloping exercises and, finally, short sprints [38] (Figure 2). Exercises were always performed barefoot and on sand. Every training session started with a 5 min warm-up program, including walking and submaximal running exercises and a dynamic stretching program. The main part of the exercise session lasted 40 min and included different types of exercises on sand [25]. The session ended with a 5 min cool-down [38]. Taken together, the duration of a single training session lasted 50 min [38]. Exercise intensity (i.e., running speed) was controlled using a stopwatch and predefined running distances on the sand-based exercise court. More detailed information on the exercise program is presented in Table 1 and Figure 2 and in a previous study by Jafarnezhadgero et al. [25]. During the intervention period, the control group performed their regular daily activities and did not perform any additional training or treatment protocols. After the post-test, individuals from the control group had the chance to receive the same training protocol as the intervention group.

### 2.3. Statistical Analyses

The normal distribution of data was confirmed through the Shapiro–Wilk test. An independent-sample *t*-test was applied to detect between-group differences at baseline. A 2 (groups: intervention, control) by 2 (time: pre, post) ANOVA for repeated measures was used to evaluate potential intervention effects. In the case of statistically significant group-by-time interactions, group-specific and Bonferroni-adjusted post-hoc tests were applied. Moreover, effect sizes were calculated by converting partial eta-squared (η2p) from ANOVA output to Cohen’s d. In accordance with Cohen [39], d < 0.50 demonstrate small effects, 0.50 ≤ d < 0.80 demonstrate medium effects, and d ≥ 0.80 demonstrate large effects. Statistical significance was set at *p* < 0.05. The Statistical Package for Social Sciences (SPSS) version 20.0 was used for all statistical analyses.

## 3. Results

All participants received treatments as allocated. Table 2 illustrates group-specific baseline data for all outcome variables. No significant between-group baseline differences were found for all examined parameters (Table 2), but the static rear foot eversion was reduced (*p* < 0.001; d = 2.83) in the intervention group at post compared with pre test (pre: 7.0 ± 0.5; post: 5.3 ± 0.7). No statistically significant pre-post differences were found for static rear foot eversion for the control group (*p* > 0.05; pre: 7.0 ± 0.8; post: 7.0 ± 0.7).

Significant main effects of “time” (*p* < 0.002; d = 0.87–0.89) and group-by-time interactions (*p* < 0.001; d = 0.94–1.18) were found for FzHC and FxHC (Table 3). Exercise resulted in significant reductions in FzHC (*p* = 0.001; d = 0.86) and FxHC (*p* < 0.001; d = 1.33) in the intervention but not the control group.

Findings demonstrated significant main effects of “time” for peak negative *free moment* amplitudes (*p* < 0.034; d = 0.56–0.58) (Table 3). In addition, significant group-by-time interactions were found for the loading rate (*p* = 0.037; d = 0.56) (Figure 3). In the intervention but not the control group, a significantly lower loading rate (*p* = 0.003; d = 0.51) was observed at post test.

With regards to electromyographic activity, results showed significant main effects of “time” for tibialis anterior and vastus lateralis activities during the loading phase (*p* < 0.049; d = 0.52–0.53) (Table 4). Moreover, significant group-by-time interactions were observed for the vastus lateralis during the loading phase (*p* < 0.019; d = 0.63) (Table 4). Training-induced increases in vastus lateralis activity were found in the intervention but not in the control group (*p* = 0.004; d = 0.67).

## 4. Discussion

This study aimed to examine the effects of exercising on sand versus control on GRFs and muscle activities during walking on stable ground in individuals with PF.

The main results of the present study were that (i) exercising on sand resulted in significant reductions in the first peak of vertical and lateral GRFs at the heel contact along with reduced loading rates; (ii) exercising on sand induced significant increases in vastus lateralis activities at the loading phase of walking in individuals with PF.

### 4.1. Effects on Walking Kinetics

Our findings on the effects of exercise on sand on walking biomechanics in individuals with PF are in agreement with the literature [40,41]. Since there is not any other study that evaluated the effects of exercise on sand on walking mechanics in adult males with PF, we decided to discuss our results in the context of studies that evaluated the effects of running exercise on walking or running biomechanics in different cohorts (e.g., healthy recreational runners [41]). For instance, a recent study reported that a six weeks exercise program (simulated barefoot running) with three sessions per week resulted in significant decreases in loading rates and impact forces in healthy adult female runners [40]. In another study, we evaluated the acute effects of running on sand vs. stable ground and not the long-term (chronic) training effects of running on sand as performed in this study [42].

In this study, running exercise on sand induced significant reductions in the first peak of vertical and lateral GRFs during heel contact, along with reduced loading rates. There is evidence that barefoot activity, particularly if conducted on unstable surfaces (i.e., sand), stimulates plantar cutaneous mechanoreceptors [43]. The enhanced afferent input may result in better pronation control and could ultimately lead to a reduction in GRFs [44]. This, however, is speculative and needs to be verified in future studies. Nevertheless, peak vertical impact GRFs and high loading rates have proven to be predictors for lower limb injuries [10]. For instance, higher loading rates and impact force values may be related to orthopedic injuries such as stress fractures [10]. Our results demonstrated that exercise on sand has the potential to lower peak impact vertical GRFs and loading rates during walking on stable ground in active male adults with PF. Therefore, the applied exercise program could have an injury preventive effect. Future studies should examine whether regular exercise on sand really reduces injury occurrence.

Besides peak vertical GRFs, medio-lateral GRFs may also contribute to injuries of the knee and hip joints [45]. A previous cross-sectional study has demonstrated that walking on sand compared with stable ground walking resulted in larger medio-lateral GRFs in healthy adult males [46]. Of note, Jafarnezhadgero and colleagues [22] could not show surface-related effects (sand vs. stable ground) while walking on peak medio-lateral GRFs in healthy controls and individuals with PF. In contrast, our results demonstrated that exercise on sand reduced peak lateral GRFs. The contrast between our findings compared with Jafarnezhadgero and colleagues [22] may be due to different study designs (cross-sectional study vs. longitudinal study) used in these two studies.

### 4.2. Effects on Muscle Activities during Walking

Our results did not demonstrate exercise-induced changes in tibialis anterior activity at the mid-stance phase of gait. It has been demonstrated that long-term training on sand did not change tibialis anterior activity at the mid-stance phase of running [25]. A previous study has shown that tibialis anterior muscle activity during loading but not during the mid-stance phase was higher in PF individuals compared with healthy ones [47,48]. This study was able to detect exercise-induced increases in vastus lateralis activity at the loading phase. Higher vastus lateralis and vastus medialis muscle activities in healthy individuals than that PF individuals during drop landing were reported in the literature [49]. Accordingly, Chang et al. [49] recommended that knee extensor muscle activation should be realized during the rehabilitation of PF individuals [49]. When taking our findings and the results of Chang [49] into consideration, it can be hypothesized that exercise on sand may have enabled participants to walk more efficiently. A recent cross-sectional study could not find any surface-related effects (sand vs. stable ground) on selected lower limb muscle activities at the loading phase in both healthy adults and individuals with PF [22]. PF is characterized by an excessive inward rotation of the foot that continues through the leg in distal-proximal sequence and ultimately results in lower limb malalignment [22]. The abnormal inward rotation of the leg may specifically affect the knee joint in as much as there is stress on the lateral facet of the patella [50]. There is evidence that PF can develop through leg length asymmetries [50]. More specifically, foot pronation can lower the ankle joint axis and may thus reduce the lower limb length slightly. Lowered arches are frequently seen in PF individuals and may put pressure on the plantar ligaments and the plantar aponeurosis (plantar fascia) [50]. If these structures are stressed over longer periods of time, micro tears, pain, and inflammation may accrue [50,51,52]. The observed training-induced increase in vastus lateralis activity could contribute to lowering the risk of injury in PF individuals by reducing the excessive inward rotation of the leg in PF individuals.

### 4.3. Strengths and Limitations

This study has some strengths and limitations that should be discussed. In terms of strengths, exercising on sand is easy to administer in many countries globally, and it is a low-cost intervention. Accordingly, large proportions of the respective populations should have access to this type of exercise. Besides its therapeutic use, exercising on sand may also contribute to fighting the pandemic of physical inactivity and sedentarism because exercising on sand is a joyful, safe (no injuries were reported in this and other studies), and effective exercise type. With regards to the study limitations, the intervention group performed the exercise training on sand while the waiting control group did not perform any additional exercise during the intervention period. It would be interesting to contrast the intervention group with an active control group performing the same exercises on stable ground in future studies. In addition, we tested young, physically active men only. Therefore, our findings are specific to this cohort and cannot necessarily be transferred to females or different age or patient groups. More research is needed in this area with different age or patient groups. In this study, we did not record kinematic data. This should be realized in future research combining kinematic, kinetic, and electromyographic data to elucidate effects and the underlying physiological mechanisms following exercising on sand. It would be worthwhile to record additional muscles such as the peroneus longus. Overall, the results of this study are useful for therapists and medical staff in general who work with PF individuals.

## 5. Conclusions

This intervention study examined the effects of 8 weeks of exercising on sand on walking kinetics and muscle activities in male individuals with PF. We found lower impact vertical and lateral peak forces and increased lower limb muscle activity (e.g., vastus lateralis) after training. Accordingly, we recommend implementing running exercises on sand as an effective treatment for individuals with PF. Further research is needed to verify whether this exercise program has the potential to reduce injury occurrence.

## Figures and Tables

**Figure 1 sports-10-00070-f001:**
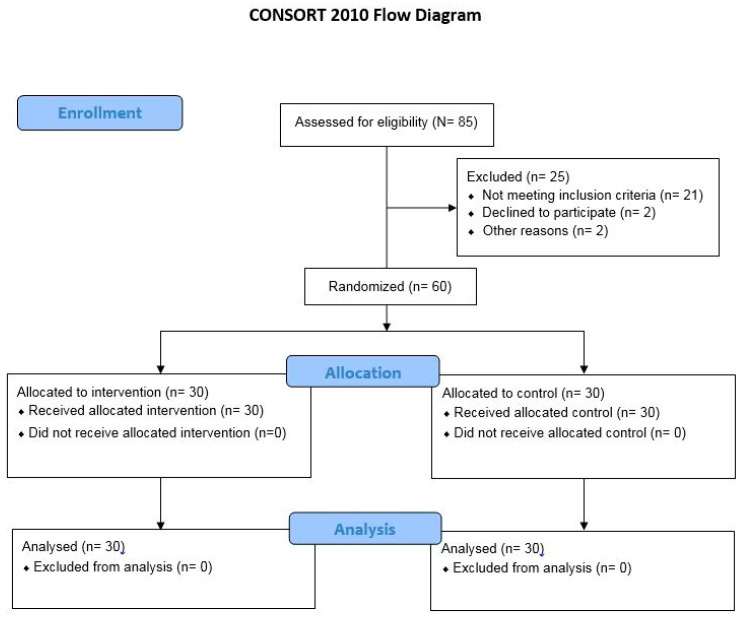
Flow diagram of the randomized controlled trial.

**Figure 2 sports-10-00070-f002:**
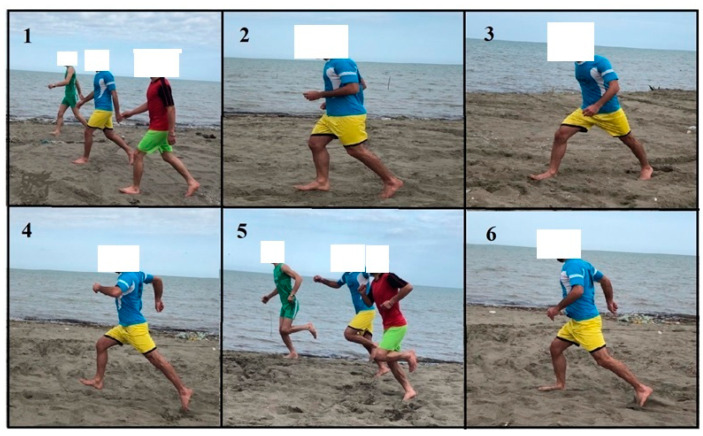
Exemplified exercises of the intervention on sand. (**1**) Walking task; (**2**) jogging task; (**3**) striding task; (**4**) bounding task; (**5**) galloping task; and (**6**) short sprints. Written informed consent was obtained from the individual for the publication of the image.

**Figure 3 sports-10-00070-f003:**
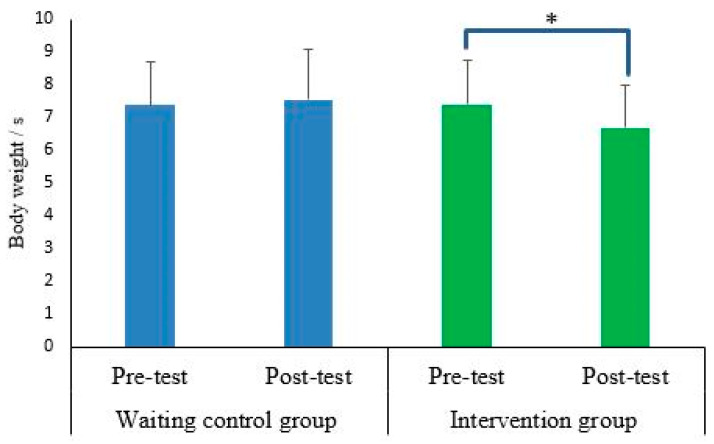
Group-specific pre-post data for loading rate (body weight/s) during walking at constant speed. Descriptive data are illustrated as means (SD). Asterisk denotes statistical significance (e.g., * *p* < 0.05).

**Table 1 sports-10-00070-t001:** Exemplified exercises on sand together with training volume and intensity of the intervention group.

Exercise	Duration (Minutes)	Intensity (m/s)		Number of Repetitions		Distance (Meters)	Rest Period (Minutes)
		First 4 Weeks	Weeks 5 to 8	First 4 Weeks	Weeks 5 to 8		
Walking	5	1.2 ±0.1	1.4 ± 0.1	-	-	50	-
Jogging	20	2.0 ± 0.1	2.5 ± 0.1	-	-	50	-
Striding	3	3.5 ± 0.2	4.5 ± 0.2	2	3	50	1
Bounding	3	3.5 ± 0.2	4.5 ± 0.2	2	3	30	1
Galloping	3	3.5 ± 0.2	4.5 ± 0.2	2	3	30	1
Short sprints	6	as fast as possible	as fast as possible	3	4–5	25	2

**Table 2 sports-10-00070-t002:** Mean (SD) baseline values for the intervention and the waiting control group for all reported outcome variables.

Anthropometrics		Waiting Control Group (*n* = 30)	Intervention Group (*n* = 30)	*p*-Value
Characteristics	Age (years)	22.23 (1.93)	22.51 (2.51)	0.955
	Height (cm)	177.92 (5.74)	178.82 (6.03)	0.869
	Mass (kg)	75.40 (7.92)	75.04 (8.21)	0.612
	Rearfoot eversion (degree)	8.4 (1.0)	8.4 (0.9)	0.935
Kinetic data				
GRFs (% Body weight)	Fz_HC_	107.38 (11.71)	106.16 (9.28)	0.657
	Fz_PO_	105.17 (9.07)	104.48 (5.87)	0.725
	Fx_HC_	11.80 (1.49)	11.51 (1.51)	0.457
	Fx_PO_	−7.21 (2.47)	−7.28 (2.32)	0.902
	Fy_HC_	−8.05 (2.44)	−8.37 (2.29)	0.601
	Fy_PO_	10.13 (4.22)	9.55 (3.24)	0.553
TTP GRFs (ms)	Fz_HC_	147.96 (19.67)	146.30 (19.36)	0.742
	Free moment (negative) × 10^−3^	−1.09 (0.44)	−1.08 (0.64)	0.946
	Free moment (positive) × 10^−3^	1.93 (0.56)	1.87 (0.43)	0.632
	Loading rate	7.39 (1.33)	7.41 (1.36)	0.956
	Stance time	0.65 (0.09)	0.66 (0.08)	0.545
Electromyographic data (%MVIC)				
Loading phase	Tibialis anterior	25.28 (6.22)	24.75 (8.12)	0.777
	Gastrocnemius medialis	7.85 (2.40)	8.19 (2.45)	0.587
	Vastus lateralis	19.41 (7.54)	19.59 (6.78)	0.923
	Vastus medialis	21.86 (7.33)	21.74 (7.64)	0.948
	Rectus femoris	21.44 (7.42)	21.61 (6.83)	0.929
	Biceps femoris	11.23 (3.88)	11.09 (4.09)	0.895
	Semitendinosus	10.94 (4.49)	11.02 (3.80)	0.885
	Gluteus medius	20.09 (6.62)	21.83 (6.74)	0.319
Mid-stance phase	Tibialis anterior	7.93 (3.26)	7.84 (3.59)	0.917
	Gastrocnemius medialis	26.26 (9.25)	26.60 (9.14)	0.886
	Vastus lateralis	6.33 (2.20)	7.02 (2.42)	0.254
	Vastus medialis	7.76 (3.22)	7.82 (2.70)	0.933
	Rectus femoris	19.29 (7.01)	19.35 (6.18)	0.971
	Biceps femoris	5.52 (2.30)	5.88 (2.07)	0.537
	Semitendinosus	6.91 (2.77)	7.01 (2.82)	0.889
	Gluteus medius	13.29 (4.22)	13.21 (5.29)	0.950
Push-off phase	Tibialis anterior	8.80 (2.35)	8.60 (2.09)	0.735
	Gastrocnemius medialis	46.19 (13.68)	46.37 (14.21)	0.961
	Vastus lateralis	6.73 (2.63)	6.40 (2.60)	0.627
	Vastus medialis	7.31 (2.12)	7.37 (2.37)	0.920
	Rectus femoris	16.66 (2.91)	16.79 (2.96)	0.865
	Biceps femoris	6.30 (2.23)	6.58 (2.05)	0.615
	Semitendinosus	6.38 (2.01)	6.59 (2.01)	0.679
	Gluteus medius	17.32 (4.28)	17.79 (4.08)	0.662

Notes: BMI, body mass index; FzHC, peak vertical force at heel contact; FzPO, peak vertical force during push-off; FyHC, braking force; FyPO, propulsion force; FxHC, peak lateral force at heel contact; FxPO, peak medial force at push-off phase; TTP, time to reach peak; x, medio-lateral orientation; y, anterior-posterior orientation; z, vertical orientation; MVIC, maximum voluntary isometric contraction; SD = standard deviation.

**Table 3 sports-10-00070-t003:** Group-specific pre-post data for ground reaction forces during walking at constant speed. Descriptive data are illustrated as means (SD) and 95% confidence intervals.

Ground Reaction Forces	Waiting Control Group (*n* = 30)				Intervention Group (*n* = 30)				Significance Level and Effect Size; *p*-Value with d-Value in Brackets		
	Pre-Test	Post-Test	95% CI	∆%	Pre-Test	Post-Test	95% CI	∆%	Time	Group	Group × Time
Fz_HC_ (% Body weight)	107.38 (11.71)	107.76 (12.84)	−3.69, 2.92	0.35	106.16 (9.28)	96.90 (8.19)	4.91, 13.61	−8.72	**0.002 (0.873)**	**0.015 (0.659)**	**0.001 (0.947)**
Fz_PO_ (% Body weight)	105.17 (9.07)	109.63 (14.60)	−9.43, 0.52	4.24	104.48 (5.87)	103.95 (7.46)	−2.34, 3.38	−0.50	0.166 (0.369)	0.137 (0.397)	0.082 (0.464)
Fx_HC_ (% Body weight)	11.80 (1.49)	12.10 (1.41)	−1.06, 0.47	2.54	11.51 (1.51)	9.41 (1.80)	1.33, 2.87	−18.24	**0.001 (0.892)**	**0.000 (1.291)**	**0.000 (1.185)**
Fx_PO_ (% Body weight)	−7.21 (2.47)	−7.31 (2.68)	−1.31, 1.52	1.38	−7.28 (2.32)	−7.65 (2.72)	−0.62, 1.36	5.08	0.576 (0.142)	0.684 (0.110)	0.757 (0.090)
Fy_HC_ (% Body weight)	−8.05 (2.44)	−8.08 (2.60)	−1.05, 1.12	0.37	−8.37 (2.29)	−8.03 (2.70)	−1.67, 0.99	−4.06	0.719 (0.090)	0.787 (0.063)	0.657 (0.110)
Fy_PO_ (% Body weight)	10.13 (4.22)	10.11 (2.84)	−1.64, 1.68	−0.19	9.55 (3.24)	9.49 (2.77)	−1.42, 1.53	−0.62	0.945 (0.000)	0.371 (0.238)	0.976 (0.000)
TTP Fz_HC_ (ms)	147.96 (19.67)	145.96 (20.09)	−5.10, 9.10	−1.35	146.30 (19.36)	148.70 (23.66)	−13.81, 9.01	1.64	0.952 (0.000)	0.900 (0.000)	0.506 (0.180)
Free Moment (negative) × 10^−3^	−1.09 (0.44)	−1.01 (0.36)	−0.27, 0.12	−7.33	−1.08 (0.64)	−0.82 (0.40)	−0.49, −0.01	−24.07	**0.034 (0.569)**	0.308 (0.271)	0.237 (0.314)
Free Moment (positive) × 10^−3^	1.93 (0.56)	1.92 (0.49)	−0.27, 0.30	−0.51	1.87 (0.43)	1.74 (0.39)	−0.09, 0.35	−6.95	0.422 (0.211)	0.154 (0.381)	0.514 (0.168)

Notes: Fz_HC,_ peak vertical force at heel contact; Fz_PO,_ peak vertical force at push-off; Fy_HC_, braking force; Fy_PO_, propulsion force; Fx_HC_, peak lateral force at heel contact; Fx_PO_, peak medial force during the push-off; TTP, time to reach peak; CI, confidence interval. Significant outcomes were highlighted in bold. 95% CI belong to pre and post values.

**Table 4 sports-10-00070-t004:** Group-specific pre-post data for muscle activities during the loading phase expressed in % of the maximum isometric voluntary contraction (MVIC). Descriptive data are illustrated as means (SD) and 95% confidence intervals.

Phase	Muscles	Waiting Control Group (*n* = 30)				Intervention Group (*n* = 30)				Significance Level and Effect Size; *p*-Value with d-Value in Brackets		
		Pre-Test	Post-Test	95% CI	∆%	Pre-Test	Post-Test	95% CI	∆%	Time	Group	Group × Time
Loading	Tibialis anterior	25.28 (6.22)	25.72 (7.15)	−3.37, 2.51	1.74	24.75 (8.12)	29.18 (9.80)	−8.30, 0.55	17.89	**0.046 (0.536)**	0.383 (0.230)	0.098 (0.439)
	Gastrocnemius medialis	7.85 (2.40)	7.93 (2.62)	−0.83, 0.67	1.01	8.19 (2.45)	7.74 (2.61)	−0.49, 1.39	−5.49	0.539 (0.168)	0.894 (0.000)	0.373 (0.238)
	Vastus lateralis	19.41 (7.54)	18.98 (5.44)	−1.92, 2.78	−2.21	19.59 (6.78)	24.32 (8.40)	−8.41, −1.05	24.43	**0.049 (0.527)**	0.071 (0.000)	**0.019 (0.633)**
	Vastus medialis	21.86 (7.33)	20.12 (5.04)	−1.41, 4.90	−7.95	21.74 (7.64)	22.27 (7.19)	−4.52, 3.45	2.43	0.628 (0.127)	0.427 (0.211)	0.363 (0.238)
	Rectus femoris	21.44 (7.42)	19.73 (6.65)	−0.10, 3.52	−7.97	21.61 (6.83)	22.08 (8.91)	−3.69, 2.75	2.17	0.495 (0.180)	0.467 (0.191)	0.233 (0.314)
	Biceps femoris	11.23 (3.88)	10.75 (3.15)	−0.88, 1.83	−4.27	11.09 (4.09)	10.66 (3.59)	−1.55, 2.41	−3.87	0.446 (0.210)	0.881 (0.000)	0.969 (0.000)
	Semitendinosus	10.94 (4.49)	10.18 (3.24)	−1.12, 2.63	−6.94	11.02 (3.80)	10.84 (3.65)	−1.69, 2.20	−2.25	0.448 (0.201)	0.581 (0.142)	0.707 (0.090)
	Gluteus medius	20.09 (6.62)	20.05 (7.04)	−2.04, 2.14	−0.19	21.83 (6.74)	19.56 (5.47)	−0.89, 5.42	−10.39	0.218 (0.327)	0.656 (0.110)	0.237 (0.314)
Mid-stance	Tibialis anterior	7.93 (3.26)	7.18 (1.96)	−0.68, 2.19	−9.45	7.84 (3.59)	9.35 (3.66)	−3.01, −0.02	19.26	0.453 (0.201)	0.115 (0.419)	**0.029 (0.586)**
	Gastrocnemius medialis	26.26 (9.25)	26.42 (7.95)	−2.60, 2.26	0.60	26.60 (9.14)	25.74 (9.33)	−3.42, 5.13	−3.23	0.777 (0.063)	0.932 (0.000)	0.673 (0.110)
	Vastus lateralis	6.33 (2.20)	6.42 (3.21)	−1.21, 1.03	1.42	7.02 (2.42)	7.73 (2.96)	−1.88, 0.46	10.11	0.319 (0.263)	0.091 (0.454)	0.436 (0.211)
	Vastus medialis	7.76 (3.22)	7.85 (4.13)	−1.47, 1.28	1.15	7.82 (2.70)	8.13 (2.80)	−1.55, 0.94	3.96	0.663 (0.110)	0.810 (0.063)	0.816 (0.063)
	Rectus femoris	19.29 (7.01)	19.40 (6.77)	−1.72, 1.50	0.57	19.35 (6.18)	18.72 (8.40)	−2.35, 3.62	−3.25	0.755 (0.090)	0.852 (0.063)	0.656 (0.110)
	Biceps femoris	5.52 (2.30)	5.84 (2.61)	−1.15, 0.51	5.79	5.88 (2.07)	6.08 (2.14)	−1.19, 0.79	3.40	0.417 (0.211)	0.562 (0.155)	0.850 (0.063)
	Semitendinosus	6.91 (2.77)	7.51 (3.55)	−2.01, 0.81	8.68	7.01 (2.82)	6.70 (2.51)	−1.13, 1.74	−4.42	0.765 (0.090)	0.546 (0.155)	0.363 (0.238)
	Gluteus medius	13.29 (4.22)	13.15 (5.80)	−1.75, 2.04	−1.05	13.21 (5.29)	14.36 (5.47)	−3.15, 0.86	8.70	0.463 (0.191)	0.630 (0.127)	0.344 (0.247)
Push-off	Tibialis anterior	8.80 (2.35)	8.04 (2.01)	−0.29, 1.80	−8.63	8.60 (2.09)	8.50 (2.44)	−0.76, 0.96	−1.16	0.204 (0.339)	0.778 (0.063)	0.327 (0.263)
	Gastrocnemius medialis	46.19 (13.68)	46.91 (15.43)	−5.09, 3.66	1.55	46.37 (14.21)	45.46 (14.40)	−5.94, 7.76	−1.96	0.961 (0.000)	0.840 (0.063)	0.683 (0.110)
	Vastus lateralis	6.73 (2.63)	6.63 (2.60)	−1.01, 1.21	−1.48	6.40 (2.60)	6.74 (2.90)	−1.37, 0.68	5.31	0.749 (0.090)	0.854 (0.063)	0.553 (0.155)
	Vastus medialis	7.31 (2.12)	7.14 (2.84)	−1.12, 1.45	−2.32	7.37 (2.37)	7.45 (3.19)	−1.32, 1.16	1.08	0.921 (0.000)	0.733 (0.090)	0.779 (0.063)
	Rectus femoris	16.66 (2.91)	16.93 (5.44)	−1.87, 1.33	1.62	16.79 (2.96)	16.94 (5.06)	−2.01, 1.71	0.89	0.729 (0.090)	0.940 (0.000)	0.921 (0.000)
	Biceps femoris	6.30 (2.23)	6.25 (2.74)	−0.89, 0.98	−0.79	6.58 (2.05)	6.95 (2.99)	−1.43, 0.68	5.62	0.641 (0.127)	0.382 (0.230)	0.549 (0.155)
	Semitendinosus	6.38 (2.01)	6.14 (1.98)	−0.74, 1.22	−3.76	6.59 (2.01)	6.26 (2.26)	−0.40, 1.06	−5.01	0.348 (0.247)	0.699 (0.110)	0.884 (0.000)
	Gluteus medius	17.32 (4.28)	17.58 (7.35)	−2.88, 2.35	1.50	17.79 (4.08)	19.12 (4.91)	−3.94, 1.28	7.47	0.382 (0.230)	0.333 (0.255)	0.558 (0.155)

Notes: PF, pronated feet; CI, confidence interval. Significant outcomes were highlighted in bold. 95% CI belong to pre and post values.

## Data Availability

Data will be available at request.

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
