# Peer review of "Eight Weeks of Exercising on Sand Has Positive Effects on Biomechanics of Walking and Muscle Activities in Individuals with Pronated Feet: A Randomized Double-Blinded Controlled Trial"

_sports, 2022, doi:10.3390/sports10050070_

Round 1

Reviewer 1 Report

General overview

The authors aimed to investigate the effects of barefoot running exercise on sand on measures of walking kinetics and muscle activities in individuals with diagnosed pronated feet.  They hypothesized that exercise on sand decreases vertical ground reaction force amplitudes,  loading rates, free moment amplitudes and increases muscle activities during walking.

In conclusion, the authors found that exercise on sand resulted in significant reductions in first peak of vertical and lateral ground reaction forces at the heel contact along with reduced loading rates; additionally, exercise on sand induced significant increases in vastus lateralis activities at the loading phase of walking in individuals with pronated feet . 

The authors did a great job. The paper is well written. Below are my comments for each section of the paper.

Specific comments

Abstract

The abstract clearly summarizes the research work and is of adequate length.
The keywords have been chosen correctly. They will help in searching for the manuscript through internet search engines.

Introduction

The authors did a good job of synthesizing the literature.
They correctly stated what is known, what is not known and formulated the Research Question / Hypothesis.
The authors clearly highlighted the gaps in the literature they sought to fill.

To fix:

  • Page 3, Lines 7-8: Replace with "Therefore, this study aimed to investigate the effects ...".

Material and Methods

The methodology used is clearly explained.
The measurements were made with reliable tools.
The study has a control group and is randomized and double-blind.
An a priori power analysis was carried out to set the minimum number of participants.
The statistics have been chosen and used correctly (Shapiro-wilk, ANOVA 2x2 with repeated measures and Cohen's d effect size).

To fix:

  • Page 3, line 27: After writing "Table 2" you should immediately insert Table 2. However, the data in Table 2 are useful in the "Results" section. I suggest changing like this: "... are described in the results section (see Table 2).

Results

The results section is written correctly. The authors described all the information from the statistical calculations in detail.

To fix:

  • Page 7, lines 15-16: Replace with "Table 2. Mean (SD) baseline values for the experimental and the waiting control group for all reported outcome variables".
  • Page 9, Table 3: Replace with "Descriptive data are illustrated as means (SD)".
  • Page 10, Figure 3: Note the meaning of the asterisk (e.g., *p <0.05)
  • Page 11, Table 4: Replace with "Descriptive data are presented as means (SD)".

Discussion and Conclusions

The authors' conclusions are justified. The take-home message is clear.
The limitations and strengths are described rightly.
The authors' findings are, in my opinion, very important because running on the sand could be an effective treatment for people with pronated feet.

Author Response

Authors’ response to the editor’s and reviewers’ comments

Manuscript ID: sports-1692905  

Eight weeks of exercising on sand has positive effects on biomechanics of walking and muscle activities in individuals with pronated feet: A randomized double blinded controlled trial

Dear Managing Editor and Reviewers,

Thank you for your careful evaluation of our manuscript and your helpful comments.

We addressed your annotations in our point-by-point statements and made changes to the manuscript whenever needed.

We hope that our revision improved the quality of our manuscript so that it is now suitable for publication in Sports.

Kind regards

The authors

Responses to Reviewer #1: changes were highlighted in yellow in the revised manuscript

Reviewer #1: Comments and Suggestions for Authors

General overview

The authors aimed to investigate the effects of barefoot running exercise on sand on measures of walking kinetics and muscle activities in individuals with diagnosed pronated feet.  They hypothesized that exercise on sand decreases vertical ground reaction force amplitudes,  loading rates, free moment amplitudes and increases muscle activities during walking.

In conclusion, the authors found that exercise on sand resulted in significant reductions in first peak of vertical and lateral ground reaction forces at the heel contact along with reduced loading rates; additionally, exercise on sand induced significant increases in vastus lateralis activities at the loading phase of walking in individuals with pronated feet . 

The authors did a great job. The paper is well written. Below are my comments for each section of the paper.

Authors’ response: Thank you very much for your affirmative comments on our paper.

Specific comments

Abstract

Reviewer #1: The abstract clearly summarizes the research work and is of adequate length.
The keywords have been chosen correctly. They will help in searching for the manuscript through internet search engines.

Authors’ response: Thank you for your positive rating of our abstract.

Introduction

Reviewer #1: The authors did a good job of synthesizing the literature.
They correctly stated what is known, what is not known and formulated the Research Question / Hypothesis.
The authors clearly highlighted the gaps in the literature they sought to fill.

Authors’ response: Thank you very much.

To fix:

Reviewer #1: Page 3, Lines 7-8: Replace with "Therefore, this study aimed to investigate the effects ...".

Authors’ response: Revised as suggested.

Material and Methods

Reviewer #1: The methodology used is clearly explained. The measurements were made with reliable tools. The study has a control group and is randomized and double-blind.
An a priori power analysis was carried out to set the minimum number of participants.
The statistics have been chosen and used correctly (Shapiro-wilk, ANOVA 2x2 with repeated measures and Cohen's d effect size).

Authors’ response: Thank you for your affirmative comments on the methods and the statistics section.

To fix:

  • Reviewer #1: Page 3, line 27: After writing "Table 2" you should immediately insert Table 2. However, the data in Table 2 are useful in the "Results" section. I suggest changing like this: "... are described in the results section (see Table 2).

Authors’ response: The reviewer is correct. Since Table 2 includes baseline data (e.g., anthropometric characteristics, kinetic data etc.), it should be removed to the results section. In accordance with the reviewer's comment, the following statement was included in the results section.

"Anthropometric characteristics of the intervention and the control group are described in Table 2."

Results

Reviewer #1: The results section is written correctly. The authors described all the information from the statistical calculations in detail.

Authors’ response: Thank you.

To fix:

  • Reviewer #1: Page 7, lines 15-16: Replace with "Table 2. Mean (SD) baseline values for the experimental and the waiting control group for all reported outcome variables".

Authors’ response: Done as suggested.

  • Reviewer #1: Page 9, Table 3: Replace with "Descriptive data are illustrated as means (SD)".

Authors’ response: Done as suggested.

  • Reviewer #1: Page 10, Figure 3: Note the meaning of the asterisk (e.g., *p<0.05)

Authors’ response: In accordance with the reviewer's comment, the following statement was added to this section:

"Asterisk denote statistical significance (e.g., *p <0.05)."

  • Reviewer #1: Page 11, Table 4: Replace with "Descriptive data are presented as means (SD)".

Authors’ response: Done as suggested.

Discussion and Conclusions

Reviewer #1: The authors' conclusions are justified. The take-home message is clear.

Authors’ response: Thank you for your affirmative comment.

Reviewer #1: The limitations and strengths are described rightly.

Authors’ response: Thank you.

Reviewer #1: The authors' findings are, in my opinion, very important because running on the sand could be an effective treatment for people with pronated feet.

Authors’ response: Thank you for your overall positive rating of our paper.

Reviewer 2 Report

Eight weeks of exercising on sand has positive effects on biomechanics of walking and muscle activities in individuals with pronated feet: A randomized double blinded controlled trial

Journal: SPORT

the aim of  this  study  was  to  investigate  the  effects  of  barefoot  running  exercise  on  sand  versus control  on  measures  of  walking  kinetics  and  muscle  activities  in  individuals  with  diagnosed pronated feet.  The authors hypothesized that exercise on sand decreases GRF amplitudes, loading rates, free moment amplitudes and increases muscle activities during walking.

The approach of the study appears very original, and the contents of the manuscript is quite interesting by his methodology.

The manuscript reads smoothly and is easy to understand. The aims, scope, and results of the study are clearly stated. I have very much enjoyed reading this paper. I find it interesting and clearly written and satisfying also all the other publication criteria of the “SPORT”. The study provides a very valuable addition to this line of research, and adds relevantly to the subject with additional original findings. I thus find that this paper definitively delivers results that will surely be of interest to the readership of the “SPORT”. I recommend the publication of this paper after minor revision: I recommend the addition of the following references that will increase the methodology and discussion sections  that appears still poor.

  • Can we modelize the movement for the better prevention? Can you develop this point LIKE a perspective of this work?

*F Fourchet ET AL (2011). Foot, ankle, and lower leg injuries in young male track and field athletes. IJATT 16 (3).

*F Fourchet,et al (2012). Comparison of plantar pressure distribution in adolescent runners at low vs. high running velocity. Gait & posture 35 (4), 685-687

Author Response

Responses to Reviewer #2: changes were highlighted in blue in the revised manuscript

Reviewer #2: Comments and Suggestions for Authors

This study was to investigate the effects of barefoot running exercise on sand versus control  on  measures  of  walking  kinetics  and  muscle  activities  in  individuals  with  diagnosed pronated feet.  The authors hypothesized that exercise on sand decreases GRF amplitudes, loading rates, free moment amplitudes and increases muscle activities during walking.

The approach of the study appears very original, and the contents of the manuscript is quite interesting by his methodology.

The manuscript reads smoothly and is easy to understand. The aims, scope, and results of the study are clearly stated. I have very much enjoyed reading this paper. I find it interesting and clearly written and satisfying also all the other publication criteria of the “SPORT”. The study provides a very valuable addition to this line of research, and adds relevantly to the subject with additional original findings. I thus find that this paper definitively delivers results that will surely be of interest to the readership of the “SPORT”. I recommend the publication of this paper after minor revision: I recommend the addition of the following references that will increase the methodology and discussion sections  that appears still poor.

  • Can we modelize the movement for the better prevention? Can you develop this point LIKE a perspective of this work?

*F Fourchet ET AL (2011). Foot, ankle, and lower leg injuries in young male track and field athletes. IJATT 16 (3).

*F Fourchet,et al (2012). Comparison of plantar pressure distribution in adolescent runners at low vs. high running velocity. Gait & posture 35 (4), 685-687

Authors’ response: Thank you very much for your overall positive comments on our paper. As suggested by the reviewer, the two studies were included as references in the discussion section. Exercising on sand is certainly an effective means to absorb ground reaction forces (e.g., lower impact vertical and lateral peaks) and increase activities of selected lower limbs muscles (e.g., vastus lateralis) when walking on stable ground. This was included in the conclusions section of this paper.
